# Association between Chronic Kidney Disease and Dynapenia in Elderly Koreans

**DOI:** 10.3390/healthcare11222976

**Published:** 2023-11-17

**Authors:** Do-Youn Lee, Sunghoon Shin

**Affiliations:** 1College of General Education, Kookmin University, Seoul 02707, Republic of Korea; triptoyoun@kookmin.ac.kr; 2Research Institute of Human Ecology, Yeungnam University, Gyeongsan 38541, Republic of Korea; 3Neuromuscular Control Laboratory, Yeungnam University, Gyeongsan 38541, Republic of Korea

**Keywords:** chronic kidney disease, dynapenia, muscle strength, sarcopenia

## Abstract

Chronic kidney disease (CKD) is caused by various factors such as chronic inflammation, oxidative stress, and obesity. Loss of muscle strength and mass is a negative prognostic factor for CKD. Therefore, in this study, we aimed to investigate the association between CKD and dynapenia in the Korean elderly. To this end, we analyzed 7029 participants from the 2014–2019 Korean National Health and Nutrition Examination Survey (KNHANES) aged ≥65 years. After adjusting for all of the covariates that could affect the results, such as physical examinations, lifestyle factors, and exercise, the association between CKD and dynapenia was found to be significant, at 1.207 (95% CI: 1.056–1.379) in CKD stage 2 and 1.790 (95% CI: 1.427–2.246) in CKD stage 3a–5. However, when sexes were analyzed separately, women were significant in both CKD stage 2 and stage 3–5 compared to normal, but only in stage 3–5 for men. Additionally, the prevalence of dynapenia increased significantly as the stage of CKD increased (normal, stage 2, and stage 3–5: 18.5%, 20.8%, and 32.3% in men and 27.5%, 34.4%, and 46.1% in women, respectively). Thus, CKD is significantly related to dynapenia, especially in women, when stratified by sex.

## 1. Introduction

The progressive disease known as chronic kidney disease (CKD) has high morbidity and mortality rate. Heterogeneous disorders affecting kidney function and structure are collectively referred to as CKD, which is classified according to disease severity levels, as evaluated by albuminuria, glomerular filtration rate (GFR), and clinical diagnosis (cause and pathology) [1,2]. The prevalence of CKD is increased by factors such as diabetes, high blood pressure, obesity, and oxidative stress [3,4]. Pathophysiological changes in CKD directly induce muscle degeneration and interfere with muscle regeneration, and CKD-induced muscle atrophy contributes to reduced exercise and physical function, a low-protein diet, and reduced functional independence [5,6].

Loss of muscle strength and mass is a negative prognostic factor for CKD [7], especially decreased muscle mass in patients with pre-dialysis CKD with deteriorated kidney function [8]. It is also associated with a lower plasma concentration of serum albumin, a higher mortality rate, and a shorter time required for the development of end-stage kidney disease [9,10,11]. Reduced muscle strength causes more severe and earlier occurrence of CKD compared to peers [12].

Dynapenia, defined as a loss of muscle strength and power in relation to muscle quality, refers to aging-related muscle weakness, not nervous system or muscle disease, and is somewhat different from sarcopenia, defined as a decrease in muscle or lean mass [13,14]. Muscle mass reduces significantly more slowly than muscle strength, suggesting that dynapenia may have a different pathophysiological mechanism than sarcopenia [15].

Moreover, the measurement of muscle strength is the main step in the sarcopenia diagnosis algorithm, and treatment for sarcopenia is recommended if there is a decrease in muscle strength, regardless of whether a decrease in muscle mass is also present [16]. Hand grip strength (HGS) is mainly used as the single indicator of muscle strength in diagnosing dynapenia and sarcopenia and has the advantage of being easy to carry out and easily measurable as a parameter for evaluating the maximum voluntary muscle strength and power [17].

Several previous studies have revealed the association between CKD and sarcopenia defined as muscle mass [15,18,19,20]. However, research on dynapenia defined as muscle strength remains insufficient, and most of the research is focused on Western countries with a small number of subjects. Additionally, variables that can affect the two diseases, such as socioeconomic factors, lifestyle habits, or exercise status, have not been considered.

Therefore, in this study, we aim to analyze the association between CKD and dynapenia in community-dwelling older adults in Korea using data from the National Health and Nutrition Survey (KNHANES), and to identify the relationship between these diseases.

## 2. Materials and Methods

### 2.1. Data Source and Sampling

The Korean Centers for Disease Control and Prevention’s KNHANES 2014–2019 data set was utilized in this study. Among adults aged ≥65 years, those who responded to both the examination and health surveys, and for whom HGS was measured, were included in this study. Among the 47,309 individuals who participated in the KNHANES, 9825 individuals who were ≥65 years old were selected. The following subjects were excluded: 1207 subjects in whom HGS was not measured; 726 subjects without data on CKD measurements; 953 subjects who had previously been diagnosed with stroke, myocardial infarction, anginal pectoris, liver cirrhosis, or cancer; and 723 non-participants in the health and nutrition survey. Finally, 7029 participants were selected (Figure 1).

### 2.2. Measurements of Variables

#### 2.2.1. Covariates

Physical examinations included height, weight, systolic and diastolic blood pressure (SBP, DBP), triglyceride, fasting glucose, body mass index (BMI), waist circumference (WC), total cholesterol, high-density lipoprotein cholesterol (HDL-C), glycated hemoglobin (HbA1c), blood urea nitrogen (BUN), and creatinine measurement variables. Blood pressure (BP) was measured using a mercury sphygmomanometer in a seated position after a 10 min rest period. Two measurements were made for all subjects at 5 min intervals, and the average of the two measurements was used for the data analyses. BMI was calculated by weight (kg)/height (m^2^). We defined the cutoff points for being underweight, normal weight, and overweight as a BMI of 18.5 kg/m^2^ or lower, 25 kg/m^2^ or lower, and 25 kg/m^2^ or higher, respectively [21]. WC was measured at the midpoint of the iliac crest’s lateral border and the bottom of the rib cage. Blood samples were collected from subjects in the morning after overnight fasting for at least 8 h, and examined at a national central laboratory. Serum glucose levels were determined by the accredited laboratory using a Hitachi Automatic Analyzer 7600 (Hitachi, Tokyo, Japan). Marital status was classified as living with or without a spouse. Individual income was divided into quartiles. Smoking status was categorized as never smokers, ex-smokers, and current smokers, and drinking status was dichotomized as current users and non-users. Frequency of resistance exercise was assessed according to participants’ answers to the question “How many times do you do resistance exercise (push-ups, sit-ups, lifting dumbbells, or barbells) a week?” The short version of the international physical activity questionnaire in Korea [22], measuring health-related physical activity, was used to measure the subjects’ current walking habits. The number of days that the subject walked ≥ 10 min at a time over the week prior was expressed. Walking was measured by total walking time in a week (TWT), calculated as follows: TWT = walking days (days/week) × walking minutes (minutes/day).

#### 2.2.2. CKD Definitions

The estimated glomerular filtration rate (eGFR), expressed as mL/min/1.73 m^2^, was determined based on the updated 2022 CKD-EPI equation to determine CKD prevalence. The CKD-EPI equation is as follows:eGFR = 142 × [(min standardized creatinine/K, 1)]α × (max standardized creatinine/K, 1) − 1.200 × 0.9938Age × 1.012 [if female]
where K is 0.7 for females and 0.9 for males, α is −0.241 for females and −0.302 for males, min indicates the minimum of creatinine/K or 1, and max indicates the maximum of creatinine/K or 1. Due to the small sample sizes, subjects with stage 3, 4, or 5 kidney function were combined into one group, leaving the following GFR groups for comparative analyses: normal and CKD 1, ≥90; CKD 2, 60–89.9; and CKD 3–5, <60 mL/min/1.73 m^2^ [23].

#### 2.2.3. Dynapenia Definitions

HGS was measured using a digital hand dynamometer (T.K.K 5401; Takei, Tokyo, Japan). The participants were asked to take a standing position with the forearm fully extended in a sideways position away from the body at the level of the thigh. The participants were instructed to grasp the device firmly for three seconds, thrice with each hand, with a minimum of 60 s of rest allowed in between each measurement [24]. The relative HGS was defined as the absolute HGS divided by BMI. The dominant HGS was defined as the maximal HGS of the dominant hand. Dynapenia was defined as the maximally measured grip strength among the six measurements (<28 kg for men, <18 kg for women) [25].

### 2.3. Data Analysis

The SPSS 27.0 Windows version (IBM, Armonk, NY, USA) was used to analyze the data. The data were weighted by reference to the complex, multistage, probability sampling design. To account for the complex survey design of stratified, random, and cluster sampling, we computed 6-year sample weights based on recommended methods from the Korea Centers for Disease Control and Prevention. Data are expressed as absolute numbers and estimated percentages (with standard errors [SE]). The χ^2^ test or student’s *t*-test was used to evaluate the differences in demographic and clinical characteristics by CKD stage and dynapenia. Multivariate logistic regression analysis was used to investigate the association of dynapenia with CKD. Multiple logistic regression analyses were used to determine odds ratios (ORs) and 95% confidence intervals (CIs). The multivariable linear regression model’s multicollinearity was determined using the variance inflation factor (VIF). A VIF greater than 10 was not accepted. The covariates used were introduced in the following order: the subjects’ general characteristics and lifestyle factors (model 1), socioeconomic indicators and exercise habits (model 2), and hematological and anthropometric factors (model 3). A *p*–value < 0.05 was considered indicative of statistical significance.

## 3. Results

Table 1 shows the characteristics of the subjects according to CKD and sex. According to the CKD stage, there was no significant difference in height, marital status, fasting glucose, and TWT in men, while there was no difference in weight, smoking status, and resistance experience in women. Additionally, there was no significant difference in individual income in men and women. The prevalence of dynapenia increased significantly as the degree of CKD increased in both men and women, as shown in Figure 2 (normal, stage 2, and stage 3a–5: 18.5%, 20.8%, and 32.3% in men, and 27.5%, 34.4%, and 46.1% in women, respectively).

Table 2 shows the characteristics of the subjects according to dynapenia and sex. For both men and women, there were no significant differences in smoking status, systolic BP, triglyceride, and HbA1C. In men, there was no significant difference in fasting glucose and TWT, whereas in women, there was no significant difference in BUN or creatine.

Table 3 shows the association between CKD stage and dynapenia. Compared to normal, the prevalence of dynapenia was 1.212 (1.135–1.295) in CKD stage 2 and 2.043 (1.672–2.496) in CKD stage 3–5. Additionally, model 3, which adjusted both variables that could affect CKD and dynapenia, showed 1.207 (1.056–1.379) in CKD stage 2 and 1.790 (1.427–2.246) in CKD stage 3–5. When analyzed separately by sex, CKD stage 2 (OR: 1.216, 95% CI: 1.024–1.444) and stage 3–5 (OR: 1.749, 95% CI: 1.292–2.369) were significant in women, but only CKD stage 3–5 was significant in men (OR: 1.791, 95% CI: 1.257–2.550).

## 4. Discussion

This study was conducted to establish the association between CKD and dynapenia in the Korean elderly. CKD and dynapenia were found to be independently associated after adjusting for various confounding variables, including socioeconomic factors, smoking, drinking, and exercise status. When classified by sex, women were significant in both CKD stage 2 and stage 3–5 compared to normal, but men were significant only in stage 3–5. Additionally, the prevalence of dynapenia increased significantly as the stage of CKD increased (normal, stage 2, and stage 3–5: 18.5%, 20.8%, and 32.3% in men, and 27.5%, 34.4%, and 46.1% in women, respectively).

The mechanism for the association between CKD and dynapenia can be explained in several ways. First, insulin resistance may have affected muscle loss in patients with CKD. Indeed, several previous studies have shown that insulin resistance and muscle strength are related [26,27,28]. In particular, insulin resistance and grip strength have a more significant correlation in women [26]. The results of these previous studies are consistent with the results of this study. In this study, the prevalence of dynapenia in women was higher (Figure 2), and the fasting glucose level was also higher in women as the CKD stage increased (Table 1). Additionally, other previous studies have suggested that insulin resistance causes muscle glucose absorption to be blunted in patients with CKD compared to the control group, which can accelerate muscle protein degradation and reduce protein synthesis, resulting in muscle wasting [29]. Therefore, insulin resistance may have contributed significantly to muscle loss in patients with CKD.

Second, an increase in IL-6 levels due to oxidative stress and chronic inflammation can decrease muscle strength and function as a detrimental result of CKD [30,31,32]. Elevations of plasma IL-6 levels and inflammatory responses are commonly observed in CKD patients, which act as triggers of the progression of CKD and related complications [31]. Additionally, CKD patients have increased levels of circulating cytokines, resulting in observed low-grade inflammation of proteins via the NF-kB pathway [30]. This promotes muscle atrophy in skeletal muscle in CKD patients and worsens muscle wasting [32].

Third, skeletal muscle undergoes apoptosis in response to renin–angiotensin system stimulation, and myostatin and apoptosis both strongly influence gene expression [33,34]. Angiotensin II administration enhances muscle proteolysis, whereas angiotensin II type 1 receptor blocker administration weakens the failure of muscle regeneration caused by transforming growth factor beta [35,36]. These results suggest that the activation of the renin–angiotensin system may affect muscle wasting in patients with CKD.

As such, pathophysiological changes at the cellular and organ system levels of CKD interfere with cellular bioenergy processes, inhibit muscle recovery and protein synthesis pathways, and increase protein degradation [32]. Resistance exercise is known to alleviate inflammation in CKD and increase muscle strength [37,38]. Our results revealed that the percentage of patients that never participate in resistance exercise according to the CKD stage was significantly higher at 75% in male patients with CKD stage 3–5, while it was not significant in women. Thus, it is thought that the relationship between CKD and dynapenia according to sex showed a significant difference only in stages 3–5 in men (Table 3), suggesting that resistance exercise is important in CKD.

Several recent studies have demonstrated the association between CKD and dynapenia or sarcopenia. In an experimental study, age and progression of CKD were found to be independent risk factors for sarcopenia, and the prevalence of sarcopenia was significantly higher in patients with CKD than in healthy participants [39]. However, this previous study excluded subjects with severe CKD symptoms, and the number of subjects was small, including only 123 patients with CKD. In another previous study, the incidence of CKD was 1.82 times higher in postmenopausal women than in healthy participants [40]. However, this previous study was disadvantaged in that there is a criterion for subjects who are postmenopausal women. Additionally, other studies have shown that the prevalence of CKD and sarcopenia defined by skeletal muscle mass measured via bioelectrical impedance analysis and dual-energy X-ray absorptiometry are related [18,41]. Although these preceding studies demonstrate clear meaningful results, unlike our research, they were defined based only on skeletal muscle mass, not muscle strength.

Despite some significant findings in this study, several limitations warrant discussion. First, although this cross-sectional study provides more information about the nature of these associations, it was a cross-sectional study that evaluated CKD and dynapenia simultaneously. Therefore, it was impossible to grasp the temporal relationship, and it was impossible to accurately grasp the order of the underlying causes between the two factors. Therefore, attention should be paid to interpreting the results, and a mechanism that can identify the causal relationship between the two factors should be identified through longitudinal studies in the future. Second, eGFR was calculated using creatinine, which is produced from muscle protein breakdown and thus depends on muscle composition. Therefore, there may be a tendency to underestimate the relationship between kidney function and muscle strength. Therefore, in future studies, it would be better to use variables that can accurately estimate kidney function while being independent of muscles, such as cysteine C. Third, the subjects who participated in the KNHANES survey may be affected by the lack of involvement of a small number of severe dynapenia or CKD patients. However, as these data were obtained from the national population, confounding factors for a small number of people are not expected to have a significant impact on the results. Furthermore, these data have the benefit of high response rates and accuracy, as determined by trained medical staff. Fourth, there is a likelihood of recall bias given that questionnaires were used to collect the socio-statistical information about the research population. However, due to the large sample size and the multistage clustered and stratified random sampling, it is unlikely that this bias had a significant impact on the study’s findings. Our results have the potential to improve the treatment of CKD and dynapenia and offer material to enhance health education in the future. Fifth, since this study did not measure the muscle mass index, patients with dynapenia according to muscle mass could not be excluded. Therefore, studies that consider both muscle mass and muscle strength will be beneficial in the future.

## 5. Conclusions

This study was conducted to establish the association between CKD and dynapenia. Although the main results of this study considered influencing variables such as smoking, drinking, and exercise, CKD and dynapenia showed an independent relationship with each other. In particular, there was a significant association between CKD and dynapenia in elderly women, whereas in elderly men, only CKD stage 3–5 was significant compared to normal. Additionally, the prevalence of dynapenia increased significantly with increasing CKD stage (normal, stage 2, and stage 3–5: 18.5%, 20.8%, and 32.3% in men, 27.5%, 34.4%, and 46.1% in women, respectively).

## Figures and Tables

**Figure 1 healthcare-11-02976-f001:**
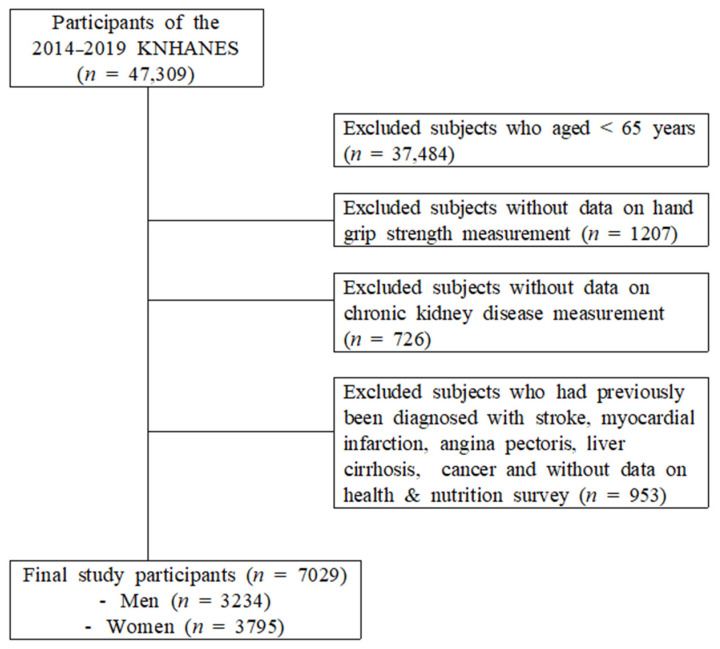
Selection of participants from the Korea National Health and Nutrition Examination Survey 2014–2019.

**Figure 2 healthcare-11-02976-f002:**
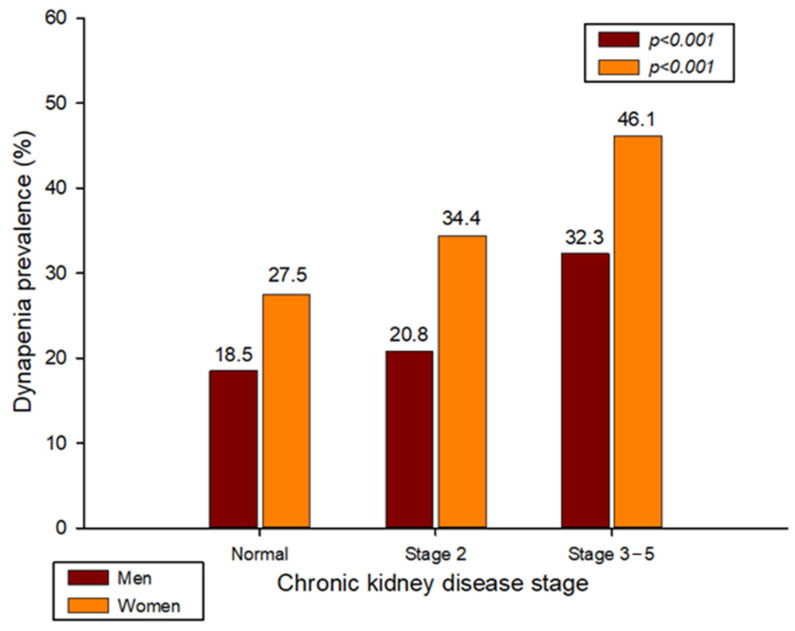
Prevalence of dynapenia according to CKD stage by sex.

**Table 1 healthcare-11-02976-t001:** Characteristics of subjects according to sex and chronic kidney disease.

	Men				Women			
	Normal (*n* = 1108)	Stage 2 (*n* = 1807)	Stage 3–5 (*n* = 319)	*p*	Normal (*n* = 1736)	Stage 2 (*n* = 1772)	Stage 3–5 (*n* = 287)	*p*
Age(y)	70.03 ± 0.15	72.82 ± 0.14	74.44 ± 0.34	<0.001	70.57 ± 0.12	74.29 ± 0.14	75.89 ± 0.32	<0.001
Height (cm)	165.67 ± 0.19	165.80 ± 0.16	165.67 ± 0.33	1.000	152.39 ± 0.17	151.77 ± 0.17	151.25 ± 0.37	0.010
Weight (kg)	64.47 ± 0.31	65.87 ± 0.25	66.74 ± 0.55	0.001	56.24 ± 0.24	56.75 ± 0.23	57.37 ± 0.57	0.114
BMI (kg/m^2^)	23.46 ± 0.11	23.93 ± 0.076	24.28 ± 0.17	0.006	24.19 ± 0.09	24.61 ± 0.086	25.05 ± 0.23	0.006
<18.5 (underweight), *n* (%)	4.0	2.7	1.7		2.0	2.0	0.5	
<25 (normal weight), *n* (%)	67.2	62.7	61.0		60.5	55.5	52.0	
≥25 (overweight), *n* (%)	28.7	34.7	37.3		47.5	42.5	37.5	
Smoking status, (%) (current/ex-/non-smoker)	22.2/56.9/20.9	16.7/62.6/20.8	19.3/60.6/20.1	0.029	2.3/2.7/95.0	2.6/3.2/94.1	2.1/5.2/92.7	0.280
Drinking status (%) (current/non-drinking)	73.6/26.4	69.7/30.3	55.3/44.7	<0.001	44.4/55.6	34.9/65.1	28.3/71.7	<0.001
Marital status (%) (living with spouse)	90.3	88.3	84.8	0.051	57.0	44.0	38.7	<0.001
Individual income				0.130				0.898
Q1 (lowest)	24.8	22.7	25.7		24.7	24.2	24.0	
Q2	26.2	22.3	24.7		24.2	24.8	28.7	
Q3	24.8	26.5	22.4		24.5	23.7	20.6	
Q4 (highest)	24.2	28.6	27.3		26.6	27.4	26.7	
SBP (mmHg)	125.53 ± 0.57	127.31 ± 0.46	128.37 ± 1.18	0.022	129.62 ± 0.48	131.45 ± 0.51	130.28 ± 1.26	0.01
DBP (mmHg)	73.08 ± 0.33	72.64 ± 0.28	68.55 ± 0.76	<0.001	73.81 ± 0.27	72.56 ± 0.27	69.30 ± 0.86	<0.001
WC (cm)	85.85 ± 0.30	87.67 ± 0.24	89.63 ± 0.53	<0.001	83.64 ± 0.26	85.09 ± 0.24	87.08 ± 0.66	<0.001
Fasting glucose (mg/dL)	108.95 ± 1.04	109.19±0.69	113.47 ± 2.07	0.111	105.43 ± 0.68	106.73 ± 0.66	113.31 ± 1.87	<0.001
Total cholesterol	178.73 ± 1.20	180.18 ± 1.10	170.6 ± 0.33	0.004	191.96 ± 1.08	190.50 ± 1.11	175.58 ± 2.45	<0.001
HDL-cholesterol	47.73 ± 0.36	46.3 ± 0.31	41.66 ± 0.63	<0.001	50.82 ± 0.33	49.56 ± 0.33	45.5 ± 0.70	<0.001
Triglyceride	131.07 ± 3.06	138.27 ± 2.51	152.67 ± 7.08	0.010	127.82 ± 2.17	135.13 ± 2.33	151.02 ± 7.97	0.010
Aerobic exercise (TWT)	66.82 ± 3.13	63.58 ± 2.42	63.21 ± 5.19	0.816	68.81 ± 2.45	56.98 ± 2.45	48.26 ± 4.77	<0.001
Resistance exercise				0.009				0.393
Never	67.8	67.0	75.0		89.1	88.9	93.6	
1–3 days/wk	12.9	9.8	8.5		4.9	5.0	2.6	
≥4 days/wk	19.4	23.3	16.6		6.0	6.1	3.8	
HbA1c	5.99 ± 0.03	6.03 ± 0.02	6.28 ± 0.06	<0.001	6.01 ± 0.03	6.07 ± 0.02	6.44 ± 0.07	<0.001
BUN (mg/dL)	15.84 ± 0.14	17.54 ± 0.12	23.63 ± 0.53	<0.001	15.58 ± 0.12	17.11 ± 0.13	23.66 ± 0.62	<0.001
Creatine (mg/dL)	0.81 ± 0.00	1.03 ± 0.003	1.52 ± 0.02	<0.001	0.63 ± 0.00	0.80 ± 0.00	1.26 ± 0.04	<0.001
eGFR (mL/min/1.73 m^2^)	94.89 ± 0.12	77.69 ± 0.22	49.84 ± 0.56	<0.001	95.17 ± 0.09	78.30 ± 0.23	48.67 ± 0.72	<0.001
Dynapenia (%)	18.5	20.8	32.3	<0.001	27.5	34.4	46.1	<0.001
Relative HGS (kg)	1.45 ± 0.011	1.41 ± 0.00	1.30 ± 0.018	<0.001	0.86 ± 0.01	0.82 ± 0.01	0.74 ± 0.02	<0.001
Dominant HGS (kg)	33.78 ± 0.24	33.31 ± 0.19	31.32 ± 0.44	<0.001	20.55 ± 0.14	19.81 ± 0.14	18.31 ± 0.34	<0.001

Data are presented as means ± SE or number (%). SBP, systolic blood pressure; DBP, diastolic blood pressure; WC, waist circumference; BMI, body mass index; TWT, total walking time in a week; BUN, blood urea nitrogen; eGFR, estimated glomerular filtration rate; HGS, hand grip strength.

**Table 2 healthcare-11-02976-t002:** Characteristics of subjects according to sex and dynapenia.

	Men			Women		
	Normal (*n* = 2526)	Dynapenia (*n* = 708)	*p*	Normal (*n* = 2613)	Dynapenia (*n* = 1182)	*p*
Age(y)	71.16 ± 0.11	75.14 ± 0.2	<0.001	71.55 ± 0.11	75.29 ± 0.15	<0.001
Height (cm)	166.43 ± 0.13	163.18 ± 0.22	<0.001	153.08 ± 0.13	149.75 ± 0.18	<0.001
Weight (kg)	66.55 ± 0.2	61.42 ± 0.39	<0.001	57.7 ± 0.19	54.18 ± 0.29	<0.001
BMI (kg/m^2^), mean (SD)	24 ± 0.06	23.05 ± 0.14	<0.001	24.61 ± 0.07	24.12 ± 0.12	<0.001
<18.5 (underweight), *n* (%)	2.1	6.4		0.9	3.9	
<25 (normal weight), *n* (%)	62.9	68.7		57.0	58.4	
≥25 (overweight), *n* (%)	35.0	24.9		42.1	37.7	
Smoking status, (%) (current/ex-/non-smoker)	18.5/61.1/20.5	20.2/58.0/21.8	0.450	2.5/2.9/94.6	2.5/3.6/94.0	0.657
Drinking status (%) (current/non-drinking)	73.9/26.1	54.0/46.0	<0.001	42.7/57.3	30.0/70.0	<0.001
Marital status (%) (living with spouse)	90.3	82.6	<0.001	53.5	40.8	<0.001
Individual income			<0.001			0.001
Q1 (lowest)	21.5	32.0		22.6	28.1	
Q2	23.4	25.6		24.1	26.2	
Q3	26.8	20.8		24.5	22.4	
Q4 (highest)	28.3	21.6		28.8	23.2	
SBP (mmHg)	126.95 ± 0.38	126.2 ± 0.78	0.381	130.19 ± 0.44	131.29 ± 0.6	0.126
DP (mmHg)	73.16 ± 0.22	69.56 ± 0.48	<0.001	73.43 ± 0.23	71.68 ± 0.34	<0.001
WC (cm)	87.62 ± 0.19	85.72 ± 0.41	<0.001	84.9 ± 0.22	83.97 ± 0.31	0.012
Fasting glucose (mg/dL)	109.83 ± 0.63	108.33 ± 1.23	0.272	106.01 ± 0.54	108.04 ± 0.89	0.049
Total cholesterol	179.64 ± 0.89	175.45 ± 1.61	0.019	191.25 ± 0.92	187.3 ± 1.29	0.012
HDL-cholesterol	46.75 ± 0.26	44.88 ± 0.47	<0.001	50.47 ± 0.27	48.42 ± 0.4	<0.001
Triglyceride	137.95 ± 2.06	134.08 ± 3.81	0.361	132.2 ± 2.08	135.06 ± 2.74	0.395
Aerobic exercise (TWT)	65.77 ± 2.12	60.58 ± 3.61	0.218	67.33 ± 2.01	49.45 ± 2.86	<0.001
Resistance exercise			<0.001			<0.001
Never	64.0	83.1		87.2	94.0	
1–3 days/wk	12.2	5.0		6.0	2.3	
≥4 days/wk	23.8	11.9		6.9	3.7	
HbA1c	6.05 ± 0.02	6.02 ± 0.04	0.564	6.05 ± 0.02	6.12 ± 0.03	0.065
BUN (mg/dL)	17.34 ± 0.1	18.22 ± 0.29	0.005	16.84 ± 0.12	17.18 ± 0.19	0.124
Creatine (mg/dL)	0.99 ± 0.01	1.02 ± 0.01	0.072	0.75 ± 0.01	0.77 ± 0.01	0.052
eGFR (mL/min/1.73 m^2^)	81.65 ± 0.33	78.72 ± 0.72	<0.001	84.57 ± 0.32	81.2 ± 0.5	<0.001
CKD stage (2/3a-5, %)	55.8/8.2	54.8/14.6	<0.001	50.8/6.3	46.1/11.2	<0.001
Relative HGS (kg)	1.51 ± 0.01	1.03 ± 0.01	<0.001	0.93 ± 0.01	0.61 ± 0.01	<0.001
Dominant HGS (kg)	35.91 ± 0.11	23.46 ± 0.2	<0.001	22.68 ± 0.07	14.43 ± 0.09	<0.001

Data are presented as means ± SE or number (%). SBP, systolic blood pressure; DBP, diastolic blood pressure; WC, waist circumference; BMI, body mass index; TWT, total walking time in a week; BUN, blood urea nitrogen; eGFR, estimated glomerular filtration rate; HGS, hand grip strength.

**Table 3 healthcare-11-02976-t003:** Odds ratios for chronic kidney disease stratified by sex according to dynapenia status.

	Crude	*p*	Model 1	*p*	Model 2	*p*	Model 3	*p*
**Total**								
CKD stage 2	1.212 (1.135–1.295)	0.002	1.280 (1.122–1.460)	<0.0001	1.215 (1.063–1.389)	0.004	1.207 (1.056–1.379)	0.006
CKD stage 3–5	2.043 (1.672–2.496)	<0.0001	2.125 (1.716–2.632)	<0.0001	1.864 (1.490–2.330)	<0.0001	1.790 (1.427–2.246)	<0.0001
**Men**								
CKD stage 2	1.157 (0.932–1.436)	0.187	1.183 (0.950–1.473)	0.132	1.190 (0.951–1.490)	0.128	1.188 (0.950–1.486)	0.132
CKD stage 3–5	2.099 (1.521–2.895)	<0.0001	1.985 (1.418–2.780)	<0.0001	1.861 (1.317–2.631)	<0.0001	1.791 (1.257–2.550)	0.001
**Women**								
CKD stage 2	1.381 (1.170–1.631)	< 0.0001	1.338 (1.132–1.582)	0.001	1.225 (1.031–1.455)	0.021	1.216 (1.024–1.444)	0.026
CKD stage 3–5	2.255 (1.697–2.997)	<0.0001	2.189 (1.630–2.941)	<0.0001	1.817 (1.338–2.467)	<0.0001	1.749 (1.292–2.369)	<0.0001

Model 1: sex, age, BMI, smoking status, drinking status. Model 2: model 1 + marital status, income, resistance exercise, aerobic exercise. Model 3: model 2 + waist circumference, blood pressure, triglyceride, HDL-C, fasting glucose. CKD reference category: normal kidney function.

## Data Availability

All data were anonymized and can be downloaded from the website (https://knhanes.kdca.go.kr/knhanes (accessed on 20 August 2023)).

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
