# Peer review of "Association between Chronic Kidney Disease and Dynapenia in Elderly Koreans"

_healthcare, 2023, doi:10.3390/healthcare11222976_

Round 1
Reviewer 1 Report
Comments and Suggestions for Authors
Thank you for the opportunity to review this manuscript. I congratulate the authors on the completion of it. It is on a topic of interest. The identification of the issues in the introduction is well documented. The innovative aspects and purpose are guided by an adequate compilation of the literature.
However, I have some comments and suggestions presented below.
Abstract
Line 15. Please remove (Model 4)
Line 18. Please, correct to: when sexes were analyzed…
Material and methods
2.2 Measurements of variables
Line 83. BMI. You present after, in the results, the BMI in categories. Please clarify according to which standards you have classified BMI?
Line 85. Please include “waist circumference” in the enumeration of Physical examinations, and explain the acronym before the line 88, where the technic of the measurement is explained.
Line 91. “BMI was calculated by weight (Kg)/height 91 (㎡).” Please move to the line 90, after the explanation of WC.
Line 90. “Blood samples were collected from subjects in the morning after overnight fasting and analyzed at a national central laboratory. BMI was calculated by weight (Kg)/height (㎡). Blood samples were obtained from each patient after fasting overnight for at least 8 h. A central and certified laboratory measured…” Please correct to: “Blood samples were collected from subjects in the morning after overnight fasting, for at least 8 h, and analyzed at a national central laboratory. The certified laboratory measured…”
Line 112. How was Scr measured?
Line 119. “2.2.2 CKD definitions” Please correct to: 2.2.3 HGS definitions
Line 125. “A resting interval of at least 60 seconds was allowed between each measurement” Please, remove it is repeated.
Line 126: “Dynapenia was defined as the maximally measured grip strength among the six measurements (<28 kg for men, <18 kg for women) [24].” Could you please, explain a little more. How you categorized dynapenia yes or not. In general you have to explain better the variables from this subheading, if not the results are difficult to understand.
Line 132. “The responses were weighted by reference to the multistage, complex, probability sampling design.” Please remove, or explain with clarity how the responses were weighted. When you read the results, it seems the responses have not been weighted.
Line 135. “Multivariate logistic regression analysis was used to investigate the association of coffee intake with sarcopenia and obesity.” Please, explain in a better way how the models were constructed. This sentence is not from this study, and you do not explain the methods you have used to select the variables you have finally included in your models: models 1, 2 and 3.
Results
Table 1 and Table 2. Please in the legend of the tables include that SCR: creatinine. I suppose. Maybe it is better to write Scr, like in the formula you have presented. To which variable do you refer with Dominant HGS.
Please move the table nearer the text that explain it.
Line 136: “between men and women”. Please change to: “in men and women”.
Table 3. Did the models show other variables with statistical significance?
Discussion
Line 187: “Additionally, other previous studies have suggested that insulin resistance causes muscle glucose absorption to be blunted in patients with CKD compared to the control group, which can suppress protein degradation, resulting in muscle wasting”. Please syntax and content review. If protein degradation is suppressed, how can it lead to muscle wasting.
Lines 208-212: “Our results revealed that the percentage of non-resistance exercise according to the CKD stage was significantly higher at 75% in male patients with CKD stage 3–5, while it was not significant in women. Thus, it is thought that the relationship between CKD and dynapenia according to sex showed a significant difference only in stages 3–5 in men (Table 3), suggesting that resistance exercise is important in CKD.” Please syntax review. You must specify that 75% corresponded to never make resistance exercise.
Line 246. “However, this procedure was probably removed at random and is unlikely to have had a significant impact on the study's findings” Please syntax review. What do you mean with removed at random?
Comments on the Quality of English LanguageModerate editing of English language required
Reviewer 2 Report
Comments and Suggestions for Authors
Dear author
I am interested in this study, however, there are several concerns before analyzing data. Thus, the author should reconsider study design and so on as follows;
Major
1) In introduction, author showed few research on the association of dynapenia, muscle strength, and physical function between CKD, however actually, there are several studies on the association. Thus, author must clearly show the novelty of this study in introduction.
2) In materials and methods, are dialysis patients included in this study? please mention.
3) Dynapenia was defined as only lower grip strength according to AWGS2019. Whereas, other studies defined dynapenia as lower grip strength and/or gait speed (lower limb function). Dynapenia in this study could exclude patients with lower gait speed. How are the author thinking about the above problem?
4) Dynapenia was defined as only measurement of grip strength, not measurement of muscle mass index. Therefore, the participants could comprise dynapenia and sarcopenia patients because of excluding low-muscle-mass patients.
Minor
5) In page 3, line 88, What is WC? Please explain it.
6) In page 4, line 119, 2.2.2 CKD definition was shown, but 2.2.3 dynapenia definition?
Sincerely
Reviewer
Round 2
Reviewer 1 Report
Comments and Suggestions for Authors
I thank you to the authors for almost all the corrections performed. Some comments presented bellow remain.
Material and methods
2.2 Measurements of variables
Line 85. Serum creatine (Scr) please change to: creatine.
Line 89. Classification of BMI. Please add a reference.
Line 113. How was Scr measured? If you used blood sample as before, please include here.
Line 128: In general, you have to explain better the variables from this subheading, for example, which variable is Dominant HGS?
Line 140. Please, explain the methods you have used to select the variables you have finally included in your models: models 1, 2 and 3.
Results
Table 1 and Table 2. Please in the legend of the tables include that Scr: creatinine.
Table 3. Did the models show other variables with statistical significance? For example, model 3, you have included WC. Is the p value associated with this covariable significant?
Discussion
Line 251. “However, this procedure was probably removed at random and is unlikely to have had a significant impact on the study's findings” Please change to: “However due to the big sample size and the multistage clustered and stratified random sampling is unlike that this bias had a significant impact on the study's findings.”
Author Response
"Please see the attachment."

Reviewer 2 Report
Comments and Suggestions for Authors
Dear author
Thank you very much for your revising.
This study failed to measure muscle mass index, so cannot exclude sarcopenia patients who have lower hand grip strength and lower muscle mass. Your study also includes lower BMI patients who have 4.0% in total, so not only dynapenia, but also probable sarcopenia or sarcopenia patients could be included. Therefore, That problem is limitation in this study, which should be mentioned as the study limitation.
Sincerely,
Reviewer
Author Response
"Please see the attachment."
